# The PpIAA5-ARF8 Module Regulates Fruit Ripening and Softening in Peach

**Yafei Qin [1], Wei Wang [1], Mingming Chang [1,2], Haiqing Yang [3], Fengrong Yin [1] and Yueping Liu [1,2,*]**

1   College of Bioscience and Resources Environment, Beijing University of Agriculture, Beijing 102206, China;
    qinyafei999@outlook.com (Y.Q.); wangwei17bj@outlook.com (W.W.); changmingming@bua.edu.cn (M.C.);
    yinfengrong1020@163.com (F.Y.)
2   Key Laboratory for Northern Urban Agriculture Ministry of Agriculture and Rural Affairs, Beijing University
    of Agriculture, Beijing 102206, China
3   Pinggu District of Fruit Bureau, Beijing 101200, China; yanghaiqing115@163.com
*   Correspondence: liuyueping@bua.edu.cn

**Abstract:** Fruit ripening and softening are important physiological processes in fruit quality formation, and auxin is involved in regulating the ripening and softening process in peach fruit. Little research has been reported on the role of Aux/IAA (auxin/indole-3-acetic acid)-ARF (auxin response factor) protein interactions in the ripening process of peach fruit. The transcriptomics and RT–qPCR results revealed that *PpIAA5* expression increased before ripening in peach fruits. Overexpression of *PpIAA5* significantly represses the expression of peach fruit ripening- and softening-related genes *PpPG* and *PpACO1* in peach fruit tissues using transient transformation. A yeast library and yeast two-hybrid screen yielded PpARF8, a protein that interacts with PpIAA5. The interaction relationship was further established using a bimolecular fluorescence complementation assay. Transient overexpression of *PpARF8* in peach fruit tissues promoted the expression of *PpPA*, *PpPG*, and *PpACO1*. Furthermore, a tomato transient transformation assay validated that the *PpARF8* gene promotes fruit ripening and softening. Taken together, our results suggest that the PpIAA5-ARF8 signaling module can affect the ripening and softening of peach fruits.

**Keywords:** peach; fruit ripening and softening; PpIAA5; PpARF8

## 1. Introduction

Peach (*Prunus persica* L.) is one of the most important cash crops in temperate regions of the world [1]. A change in the texture of peach fruit, with a decrease in hardness and fruit softening, means that the peach fruit is entering the ripening stage. The partial or complete dissolution of pectin and cellulose in the cell wall and the cleavage of starch and other polysaccharides are responsible for the change in fruit texture [2–5]. Fruit ripening is an important physiological process in the formation of fruit quality and has been of great interest to geneticists and breeders.

Several plant growth regulators (PGR), including auxin, ethylene, and abscisic acid, are involved in the peach fruit ripening process [6]. Recent studies have shown that auxin plays an important role in regulating fruit ripening [7–9]. The application of exogenous auxin treatments to strawberries, grapes, and tomatoes during the preripening period was able to inhibit fruit ripening [8,10]. However, auxin treatment applied to apples and pears before fruit ripening can induce ethylene synthesis to promote fruit softening and ripening [11–13]. In many tomato ripening mutants, endogenous auxin levels are much higher than those in normal fruits, and many studies have shown that a decrease in auxin content is one of the key factors in initiating fruit ripening [14,15]. These findings reveal a dual role of auxin in regulating fruit ripening and softening.

The most important form of auxin in plants is indole-3-acetic acid (Aux/IAA) [16]. The typical auxin response pathway relies on the interaction between the C-terminal structural

domain of the transcription factor Aux/IAA and ARF proteins [17]. The transcriptional activation or repression of ARFs is determined using a nonconserved intermediate domain, and the conserved B3 DNA-binding domain at the N-terminal end can bind to the auxin response element of the auxin response gene promoter [17,18]. Aux/IAA negatively regulates the activity of ARFs via protein–protein interactions, and ARF proteins further alter the expression of downstream genes [19,20]. Thus, the Aux/IAA-ARF module of the plant is a key component that enables the auxin signaling pathway [21].

*Aux/IAA* genes are a multigene family involved in the process of fruit ripening. In tomatoes, the SlARF7/SlIAA9 complex can control tomato fruit formation by repressing the transcription of feedback-regulated genes [22]. PpIAA19 overexpression in tomatoes is involved in regulating fruit shape [23]. In peaches, PpIAA1 is a positive regulator that accelerates fruit ripening and softening by promoting the expression of ethylene synthesis and ripening-related genes [24]. In apples, the expression of anthocyanin synthesis genes decreased in auxin-treated calli, and MdIAA26 overexpression reduced the inhibitory effect, which confirmed that *Aux/IAA* genes affected fruit coloration [25]. The application of exogenous auxin to tomatoes increased the accumulation of phenolic volatiles and altered the expression of numerous key genes involved in the aroma volatile biosynthesis pathway [26]. These results imply that auxin signaling-related transcription factors are involved in the ripening process, including fruit size, softness, color, and aroma [27], and studies related to their regulation of fruit ripening and softening have been of interest to scholars.

The functions of the ARF gene family have been studied in different physiological processes, among which the ARF gene family members related to fruit development and ripening regulation have been most studied. Unisexual tomato fruits with silenced SlARF5 had less ventricular tissue development and lighter fruit size and weight, suggesting that ARF regulates early fruit set and development in tomatoes [28]. SlARF2 has a positive regulatory role in tomato fruit ripening [29,30]. It has also been reported that enhanced expression of the auxin signaling gene SlARF10A/10B/17 affected the respiration rate, increased fruit hardness, and reduced fruit weight, thereby delaying tomato fruit ripening [31]. In apples, MdARF5 was found to induce ethylene synthesis by directly promoting the expression of the ethylene synthesis genes MdACS3a, MdACS1, and MdACO1, which further promoted apple fruit ripening [32], and it has been shown that ARFs can act as positive or negative regulators during fruit development and ripening. However, there are few studies on ARF regulation of peach fruit ripening and softening, so it is crucial to study its function.

In our previous study, 23 *PpIAAs* were identified in peach fruit, in which the expression levels of *PpIAA1/5/9* were positively correlated with the degree of fruit softening [33]. Moreover, 18 *PpARFs* were identified in peach fruit, the expression of some genes correlated with the fruit ripening process, and a dual relationship between some PpARF and PpIAA factors was confirmed [34]. These findings suggest that the transcription factors Aux/IAA and ARF may be involved in the peach fruit ripening process, but few studies have reported the specific factors involved and their regulatory mechanisms.

In this study, we further explored the Aux/IAA and ARF factors related to peach fruit ripening and found that the level of PpIAA5 (Prupe.3G074800) transcription was closely related to the fruit ripening process, and we identified PpARF factors that interacted with PpIAA5 using a protein–protein interaction experimental technique. PpARF8 (Prupe.3G182900) was selected for further investigation, and its possible downstream target genes were explored. In conclusion, we found that the PpIAA5-ARF8 module participates in the ripening and softening process in peach fruit. Our results lay a theoretical foundation for elucidating the mechanism by which auxin signal transduction factors regulate ripening and softening in peach fruits.

## 2. Materials and Methods

### 2.1. Plant Material

In 2021, 'Beijing No. 24' (Jingyan) peach fruits were harvested in the orchard of Houbegong Village, Dahuashan Town, Pinggu District, Beijing, China. The peach fruits were sampled at five developmental stages: 111 days after full bloom (early second rapid growth period, S3-1), 118 days after full bloom (late second rapid growth period, S3-2), 125 days after full bloom (early ripening period, S4-1), 132 days after full bloom (middle ripening period, S4-2), and 139 days after full bloom (fully ripening period, S4-3). The mesocarp discs were rapidly separated and immediately frozen in liquid nitrogen and then stored at $-80$ °C for RNA-seq sequencing and RT–qPCR.

The tobacco material *Nicotiana benthamiana* was grown in a culture room under 16/8 h light and dark cycle culture conditions, with relative humidity controlled at 70% and temperature conditions at 25 °C. It was used for infiltration when it grew to roughly five or six leaves per plant after 5–6 weeks.

### 2.2. Extraction of RNA and Quantitative Real-Time PCR (RT–qPCR)

Total RNA was extracted using the EASYspin Plant RNA Rapid Extraction Kit (RA106-02, Biomed, Beijing, China). A NanoDrop Lite ultraviolet spectrophotometer (ALLSHENG Nano300, Hangzhou, China) was used to estimate the quality of total RNA. The final concentration of each sample was adjusted to 1000 ng/μL (Liscum and Reed, 2002). The TransScript® First-Strand cDNA Synthesis SuperMix kit (TransGen Biotech, Beijing, China) was used to synthesize first-strand cDNA. Quantitative reverse transcriptase-PCR (qRT–PCR) analysis was conducted using a Real-Time PCR System (QuantStudio™6 Flex System, Thermo Fisher Scientific, Waltham, MA, USA), with a total reaction volume of 20 μL (9 μL SYBR Premix Ex Taq II, 1 μL For-Primer, 1 μL Rev-Primer, 2 μL cDNA, 6.6 μL ddH$_2$O, 0.4 μL Rox I). The RT–qPCR program was as follows: 95 °C for 3 min, followed by 40 cycles of 95 °C for 5 s, 55 °C for 10 s, and then 72 °C for 30 s, followed by a continuous increase in temperature from 60 °C to 95 °C for melting curve analysis. The internal reference gene was the *Translation Elongation Factor2* (TEF2) of peach [35]. The cycle threshold (Ct) $2^{-\Delta\Delta Ct}$ method was used to estimate the relative gene expression levels. Each sample consisted of three biological replicates. The primer sequences used for RT–qPCR are listed in Table S1. All amplified fragments are between 100–200 bp.

### 2.3. Vector Construction

The PCR primer sequences involved in vector construction are listed in Table S2. Genomic DNA or cDNA generated from different developmental stages of peach mesocarp were used as templates for sequence amplification. For overexpression, the full-length coding sequences of OE-*PpIAA5* (Prupe.3G074800) and OE-*PpARF8* (Prupe.3G182900) were cloned and recombined into the pCAMBIA3301-121 vector (Biomed, Beijing, China). For the construction of the vector pTRV2-*PpARF8*, a 360-bp fragment of *PpARF8* was amplified and inserted into the vector pTRV2 (HonorGene, Changsha, China). To construct a vector with a GFP tag for subcellular localization experiments, we amplified the CDS fragment of PpIAA5/ARF8 and inserted it into the vector pBI121-GFP (HonorGene, Changsha, China). For bimolecular fluorescence experiments, the full-length coding sequences of the *PpIAA5* and *PpARF8* genes without stop codons were amplified and inserted into the pSPYNE173 and pSPYCE(M) vectors (TIANGEN, Beijing, China), respectively.

### 2.4. Subcellular Localization Analysis

The full-length coding sequence of the *PpIAA5/ARF8* gene without a stop codon was amplified and inserted into the pBI121-GFP vector by a seamless cloning method. *PpIAA5/ARF8-GFP* was transiently expressed in tobacco leaves by *Agrobacterium* infiltration (GV3101) [36]. Three days after injection, the yellow fluorescent signal was detected using a laser confocal microscope (LEICATCS SP8, Weztlar, Germany) at 514 nm.

## 2.5. Transient Analysis of Peach Fruit Tissues

The successful OE-*PpIAA5* and OE-*PpARF8* plasmids were transformed into *Agrobacterium tumefaciens* GV3101. A single colony of *Agrobacterium* cells was inoculated in 15 mL of liquid LB medium with appropriate antibiotics until $A_{600}$ reached approximately 0.8–1.0. After centrifugation, the cells were resuspended in buffer (10 mM MES, 10 mM $MgCl_2$, 100 mM acetosyringone, pH 5.6) and infiltrated into peach fruit tissues at the S4-1 stage. Transient expression treatments were performed with five biological replicates.

## 2.6. Yeast Library Screening

After self-activation detection of PpIAA5, the pGBKT7-PpIAA5 (EK-Bioscience, Shanghai, China) plasmid was transformed into yeast strain Y187 and positive yeast colonies were picked and cultured in SD/-Trp liquid medium until the $A_{600}$ was 0.8–1.2. The supernatant was removed using centrifugation, resuspended in 2 × YPDA medium containing 3 mL of yeast library secondary bacteria, and incubated at 30 °C for 1 day. The supernatant was discarded using centrifugation, and the precipitate was resuspended in 0.5 × YPDA medium and spread on 40 SD/-Trp-Leu-His-Ade plates. Next, the bacteria were picked into 0.9% NaCl solution and dropped on SD/-Trp-Leu-His-Ade+AbAi+X-α-gal medium for screening, and the colonies that turned blue were amplified. The yeast plasmid DNA was extracted and transferred into DH5α for sequencing. The sequencing data were analyzed on the NCBI (https://blast.ncbi.nlm.nih.gov/Blast.cgi) database on 11 May 2022.

## 2.7. Yeast Two-Hybrid Assay

The primer sequences for the construction of the yeast two-hybrid vector are shown in Table S3. The recombinant plasmids PpIAA5 and PpARF2/3/4/6/7/8/19 were co-transformed into AH109 yeast receptor cells, and the bacterial solution was coated in SD/-Trp-Leu solid medium and incubated at 28 °C for 48–72 h. Single colonies were picked and resuspended in sterile water, and 5 µL of the bacterial solution was dropped onto SD/-Trp-Leu-Ade-His solid medium with a concentration of 100 mg/L X-α-gal. Trp-Leu-Ade-His solid medium with a concentration of 100 mg/L X-α-gal and incubated at 28 °C for 24 h in the dark to observe the growth and changes in color of the colonies.

## 2.8. Bimolecular Fluorescence Complementation Experiment

The full-length coding sequences of the *PpIAA5* and *PpARF8* genes without stop codons were amplified using PCR and inserted into the pSPYNE173 and pSPYCE (M) vectors using sticky end ligation or seamless cloning. The recombinant plasmids were transferred into *Agrobacterium tumefaciens* strain GV3101. Positive *Agrobacterium* strains were transferred to an LB liquid medium with the appropriate antibiotics and incubated at 28 °C until the $A_{600}$ reached 0.5–0.6. After centrifugation, the cells were resuspended in buffer (10 mM MES, 10 mM $MgCl_2$, 100 mM acetosyringone, pH 5.6) of $A_{600}$ exactly 1.0 and were infiltrated into the leaves of 4- to 6-week-old tobacco plants. Three days after injection, the yellow fluorescent signal was detected using a laser confocal microscope (LEICATCS SP8, Weztlar, Germany) at 514 nm [37].

## 2.9. Transient Transformation and Phenotypic Analysis of Tomato Fruits

The successful OE-*PpARF8* and pTRV2-*PpARF8* plasmids were transformed into *Agrobacterium tumefaciens* GV3101. The pTRV is a bipartite virus used in Virus-Induced Gene Silencing (VIGS). This method involves two Agrobacterium tumefaciens strains: one with the pTRV1 plasmid for viral replication and movement and the other with the pTRV2 plasmid, containing the coat protein gene and VIGS-inducing sequence. Tomato fruits were transformed by collecting *Agrobacterium* and resuspending them in buffer (10 mM $MgCl_2$, 10 mM MES, pH 5.6, 100 mM acetosyringone) and shaking for 3–4 h at room temperature [38]. The $A_{600}$ of *Agrobacterium* was adjusted to 0.8 with permeation buffer and then injected into tomato fruits at the breaker stage. The *Agrobacterium* injection site for transient transformation in toma-

toes is the peduncle. Transient expression treatments were performed with three biological replicates. The color phenotype was taken one week after infiltration.

### 2.10. Statistical Analysis

All experiments were performed at least three times. All qRT-PCRs and other quantitative analyses were repeated at least three times. The statistical data is presented with the significance levels indicated by asterisks. Specifically, * $p < 0.05$ and ** $p < 0.01$ are used to denote the level of significance in the results. Standard deviations are within a range of approximately $\pm 0.05$. Student's *t*-test was used to evaluate significant differences.

## 3. Results

### 3.1. Expression Analysis and Subcellular Localization of PpIAA5

By transcriptome analysis and identification, it was observed that the relative expression of *PpIAA5* from Figure 1a increased during the S4-1 period, and the results obtained using RT–qPCR were consistent with the transcriptome results. The transcriptome data used are all from Tables S4–S7. PpIAA5 was inferred to be closely related to the fruit ripening process. The localization of PpIAA5 was predicted using the predicted protein online software, and both the prediction results and subcellular localization experiments indicated that the protein encoded by the peach PpIAA5 gene is located in the nucleus (Figure 1b) and is tissue-specific.

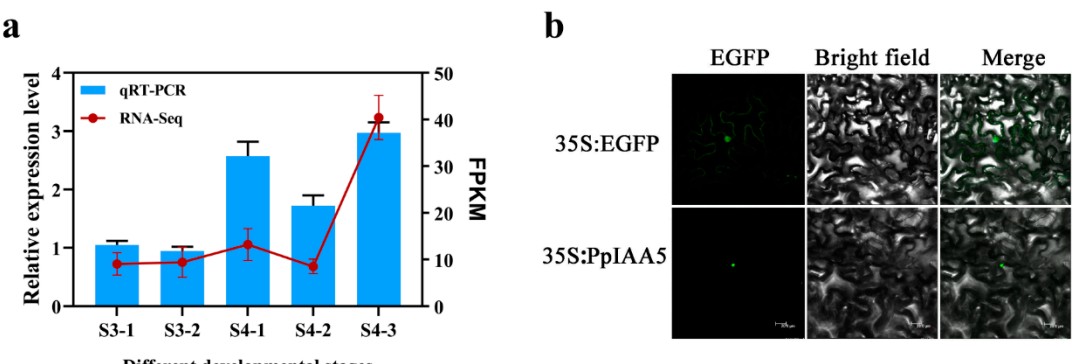

**Figure 1.** Expression analysis and subcellular localization of PpIAA5. (**a**) Relative expression level of PpIAA5 during peach fruit ripening (Note: S3-1, early second rapid growth period; S3-2, late second rapid growth period; S4-1, early ripening period; S4-2, middle ripening period; S4-3, fully ripening period.); (**b**) The subcellular localization results showed that PpIAA5 was localized in the nucleus, Scale bar = 20 μm.

### 3.2. PpIAA5 May Be a Negative Regulator in the Softening Process of Fruit Ripening

Studies at the transcriptional level revealed that *PpIAA5* may be associated with peach fruit ripening and softening, so the function of the *PpIAA5* gene was further investigated. OE-*PpIAA5* was transiently expressed in isolated peach fruit tissues. The results are shown in Figure 2. After transient overexpression of the *PpIAA5* gene, the expression of the *PpIAA5* gene was 3-fold higher than that in the control. The expression of the *PpIAA11/17/29* gene was 2-fold lower than that of the control. The expression level of the *PpARF5/7/8/16/18* gene was 2-fold lower than that of the control. The gene expression levels of *PpSAUR50*, *PpGH3.1*, and *PpYUCCA2/6/10* related to auxin signal transduction and synthesis were significantly lower than those of the control group, with the gene expression of *PpYUCCA6/10* being more than 10 times lower than that of the control group. The expression level of the *PpSAM* gene related to ethylene signal transduction was six times higher than that of the control group, and the expression level of *PpERF4/034* was not significantly different from that of the control group. Overexpression of *PpIAA5* affected the expression of some cell wall degradation-related enzyme levels in the fruit tissues, in which the expression of *PpEXP2*,

*PpMEI*, *Ppβ-GAL*, and *PpGLU* was 3-fold higher than that of the control. The expression levels of *PpPG* and *PpACO1* were 10 times lower than those of the control group.

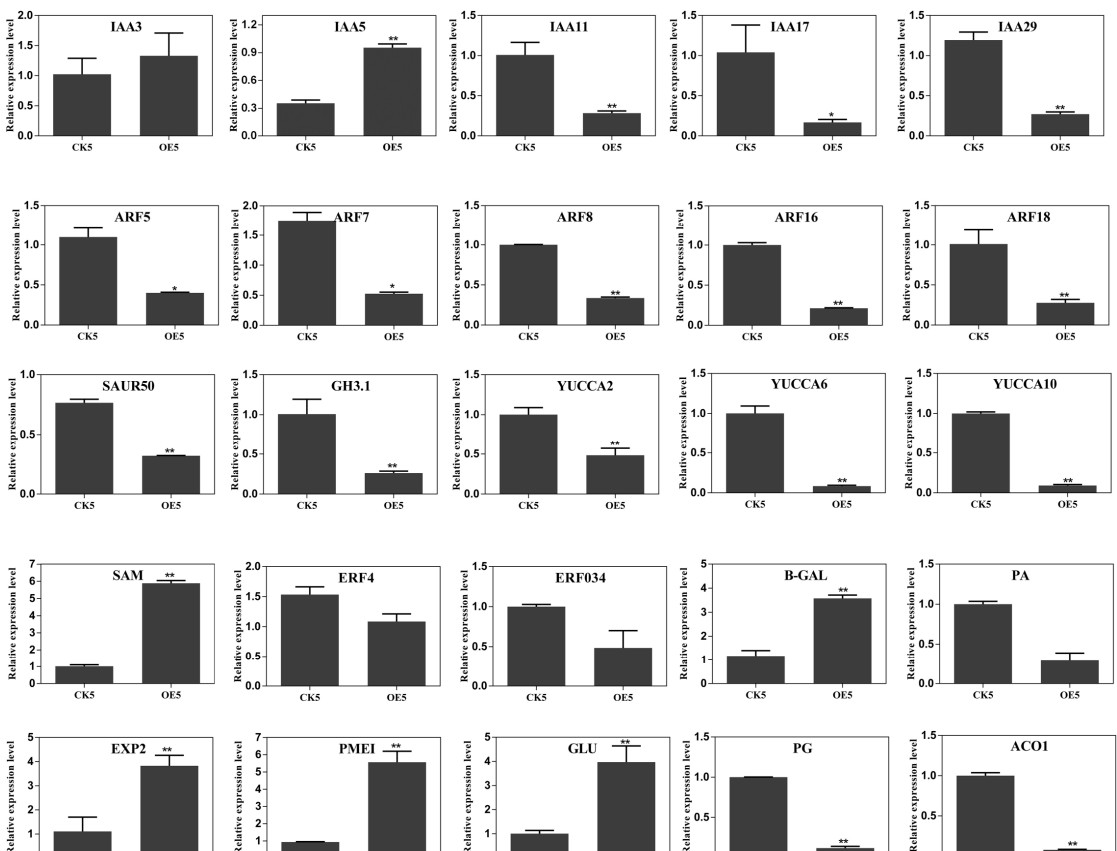

**Figure 2.** The effect of PpIAA5 overexpression in isolated peach tissues on the relative expression of related genes (Note: CK5, empty vector control; OE5, overexpression of the *PpIAA5* gene; * $p < 0.05$, ** $p < 0.01$).

### 3.3. PpIAA5 Interacted with PpARF8 In Vivo

To confirm that PpIAA5 can bind different protein factors to regulate related physiological processes, the decoy protein of PpIAA5 was used for yeast library screening (Figure 3a), and the sequencing results were analyzed using BLAST. A total of six proteins were screened for interactions with PpIAA5, including PpARF5, PpARF8, and PpARF19.

To further verify the interaction between PpARFs and PpIAAs in peach fruit, a yeast two-hybrid assay was conducted and revealed that several PpARF and PpIAA factors in peach fruit interacted physically. Among them, PpARF6/7/8/19 interacted with PpIAA5, and the strongest interaction effect was with PpARF8 (Figure 3b). After constructing a fluorescent bimolecular complementary vector of PpARF8 and PpIAA5 and transiently transforming tobacco leaves, yellow fluorescence was visible in the nuclei of tobacco cells cotransformed by PpARF8 and PpIAA5, which demonstrated that PpARF8 interacted with PpIAA5 further (Figure 3c).

### 3.4. PpARF8 Is Involved in the Regulation of the Ripening and Softening of Peach Fruit

A phylogenetic tree was built using MEGA 5.0 software to analyze the affinities of PpARF8 among different species. Figure S1a shows that *PpARF8* has high homology with proteins in lentils, apricots, apples, moonflowers, strawberries, pears, mulberries, white pears, and dates. Among them, *PdARF6-like* (lentil) and *PaARF6-like* (apricot) both belong to the Rosaceae family, and these two genes have the highest homology with *PpARF8*.

The protein encoded by the peach PpARF8 gene is located in the nucleus and has tissue specificity (Figure S1b).

**Figure 3.** Screening of PpIAA5-interacting proteins. (**a**) PpIAA5 yeast library screening, SD/−Trp−Leu−His−Ade+AbAi+ Growth on an X-α-gal plate indicating a protein interaction with PpIAA5 (Note: +: Co−transformation of pGBKT7−53/pGADT7−T was positive control, −: Cotransformation of pGBKT7−lam/pGADT7−T was negative control.); (**b**) The interaction between PpIAA5 and PpARFs was verified by yeast two-hybrid; (**c**) The interaction between PpIAA5 and PpARF8 was verified using bimolecular fluorescence complementation experiment, Scale bar = 20 μm.

To further investigate the function of the *PpARF8* gene, OE-*PpARF8* was transiently expressed in isolated peach fruit tissues. The expression of the *PpARF8* gene, genes related to auxin signaling, and cell wall degradation-related enzyme genes associated with fruit softening were later determined using RT–qPCR, and the results are shown in Figure 4. After transient overexpression of the *PpARF8* gene, the expression of the *PpARF8* gene was significantly increased by more than 12-fold. The expression levels of *PpIAA3/5* and *PpARF5/16/18* were significantly higher than those of the control group. *PpIAA11/17/29* and *PpARF7* did not show obvious changes. After overexpression of *PpARF8* compared to the control, the gene expression of *PpIAA5* was more than 1000 times that of the control. The expression of genes related to fruit ripening showed different expression profiles. The changes in the *PpERF4* and *PpERF034* genes were not significant compared to the control group. The *PpSAM/EXP2/β-GAL* genes showed no significant changes compared to the control group. The expression level of the *PpACO1* gene was 4-fold higher than that of the control group. The gene expression of *PpPMEI* was lower than that of the control group, and the expression of *PpGLU, PpPA,* and *PpPG* was higher than that of the control group.

### 3.5. PpARF8 Enhances Fruit Ripe Ning and Softening in Tomato

The transient transformation system of tomato fruits was used to transiently transform silenced and overexpressed *Agrobacterium PpARF8* into tomato peduncle in the green ripening stage, and the gene function of *PpARF8* was further characterized by phenotypic observation of tomato fruits. Compared with the control, tomato fruits ripened earlier after overexpression of the *PpARF8* gene (Figure 5c), and significant early ripening was observed in the *Agrobacterium* infestation range. However, significantly delayed ripening was observed in the *Agrobacterium* infestation range after gene silencing of *PpARF8* (Figure 5f).

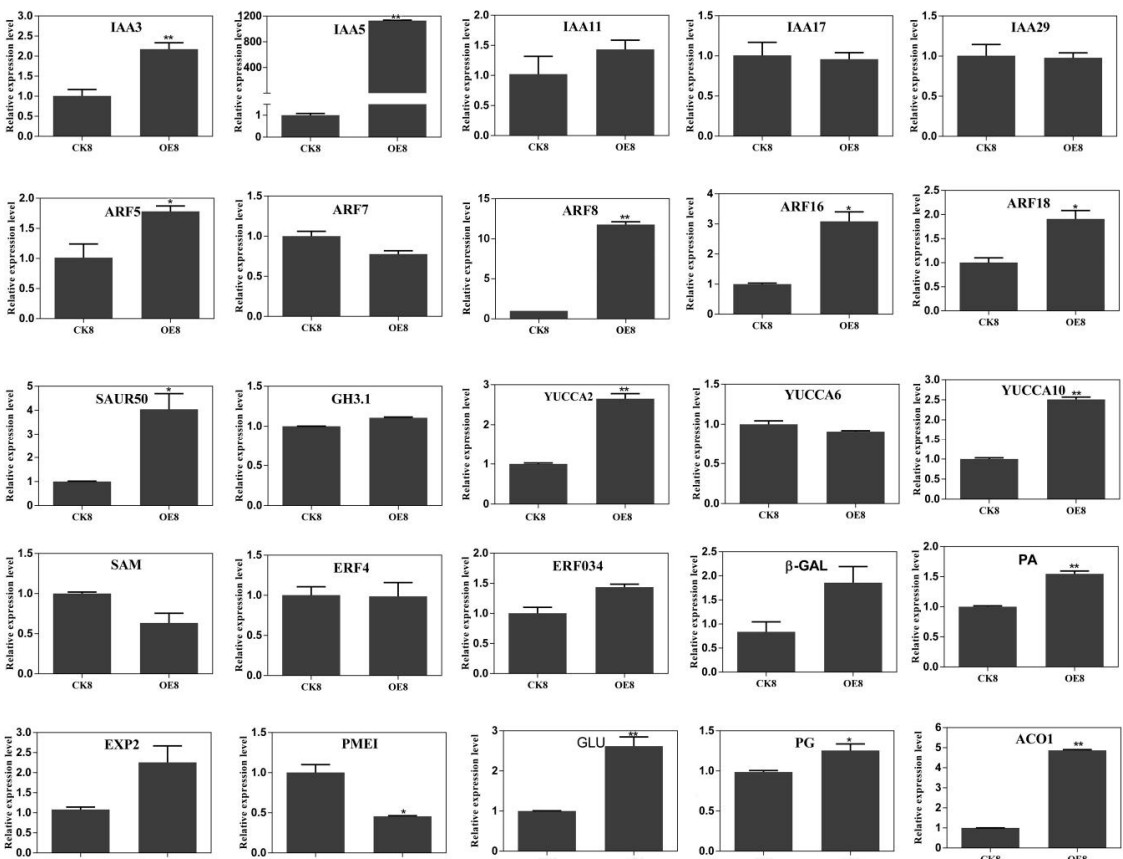

**Figure 4.** Effect of overexpression of the PpARF8 gene in isolated peach tissues on related gene expression (Note: CK8, empty vector control; OE8, overexpression of the *PpARF8* gene; * *p* < 0.05, ** *p* < 0.01).

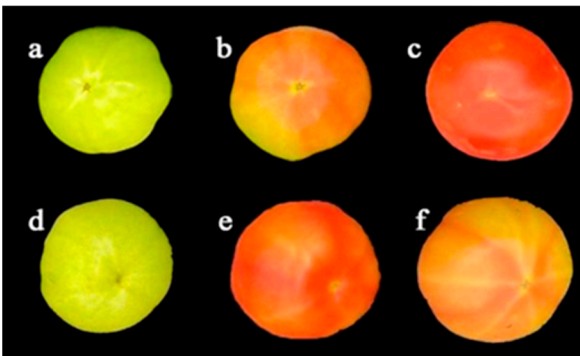

**Figure 5.** Phenotype of Agrobacterium-mediated PpARF8 gene silencing and overexpression in tomato fruit. (**a**,**d**) green mature tomato; (**b**) tomato transiently expressing pCAMBIA3301-121; (**c**) tomato transiently expressing pCAMBIA3301-121-PpARF8; (**e**) tomato transiently transformed with pTRV1 and pTRV2; (**f**) Tomato transiently transformed with pTRV1 and pTRV2-PpARF8.

## 4. Discussion

The decrease in fruit hardness and fruit coloration are signs of peach fruit ripening. Disruption of the pectin-cellulose-hemicellulose (P-C-H) structure is the essential cause of fruit ripening and softening [4]. The inhibition of *PpACS1* expression in hard peaches has been shown to be the result of low auxin concentrations. However, after 1-naphthylacetic acid (NAA) treatment, higher auxin levels led to an increase in *PpACS1* expression, as well as a softening of the peach fruit [4,5]. In previous studies, it was demonstrated that the sudden increase in auxin concentration before peach fruit ripening was accompanied by a

jump in ethylene [24], which was the reason for auxin-regulated ripening and softening in peach fruit.

Our preliminary work has confirmed that auxin is involved in the ripening and softening process of peach fruit. Generally, auxin completes the regulation of plant physiological processes via typical TIR1/AFB-Aux/IAA-ARF. Aux/IAA proteins are critical for auxin-mediated developmental signal transduction, and their function in peach fruit ripening and softening is one of the key questions to be analyzed in this experiment. Many *Aux/IAA* genes have been identified and analyzed in the developmental ripening process of fruits. Inhibition of SlIAA3 gene expression in tomatoes causes physiological characteristics of auxin- or ethylene-related developmental defects [39]. In our previous study, through transcriptome sequencing, 15 genes related to auxin signal transduction and 10 genes related to peach fruit ripening were screened. Furthermore, by RT–qPCR, five *Aux/IAA* genes more closely related to the ripening and softening process of peach fruit were obtained. Among them, the expression of *PpIAA5* was significantly higher at the S4-1 stage, which was similar to the transcriptome results (Figure 1a). Based on the transcription level results, it is speculated that *PpIAA5* may be involved in the ripening and softening process of peach fruits.

Some findings suggest that the *Aux/IAA* gene regulates peach fruit ripening using promoting ethylene synthesis and the expression of ripening-related genes [24]. In this experiment, the relationship between *PpIAA5* and peach fruit ripening was explored by transiently overexpressing the *PpIAA5* gene in peach fruit tissues and analyzing the changes in the expression of auxin-responsive genes as well as genes related to fruit ripening. The experimental results implied that the *PpIAA5* gene might affect fruit development and ripening softening by regulating downstream *PpARF* genes, including *PpARF5/7/8/16/18* expression levels that were 2-fold lower than those of the control group. Overexpressed *PpIAA5* decreased the expression of the peach fruit ripening softening-related genes *PpPG* and *PpACO1* (Figure 2), which suggests that PpIAA5 might be a suppressor. Other transcription factors may be involved in coregulating the process.

In the auxin signaling pathway, high auxin levels activate Aux/IAA protein degradation via the 26S proteasome. ARFs are thus released to upregulate downstream auxin response genes [40]. The interaction between ARF and Aux/IAA proteins is an important biochemical process in response to the auxin response [19,41]. In this study, we used yeast library screening technology and yeast two-hybrid assays to screen PpARF8, which has the strongest interaction effect with PpIAA5. The bimolecular fluorescence complementation assay further verified their interaction (Figure 3). The results were similar to those of previous studies: SlIAA9-SlARF7 controlled tomato fruit formation [22], and the MdIAA121-MdARF13 model regulated anthocyanin biosynthesis in apple fruit [42]. These results demonstrated the involvement of Aux/IAA protein dimerization with ARF protein, resulting in effects on plant growth and development, among others. Our experiment preliminarily showed that the PpIAA5-PpARF8 model may be involved in the ripening process of peach fruit.

To explore the function of PpARF8 in fruit ripening and the possible regulatory mechanism, the differentially expressed *PpARF* genes were analyzed based on transcriptome data. *PpARF5*, *PpARF7*, *PpARF8*, *PpARF16*, and *PpARF18* may play crucial roles in fruit development and ripening, given their higher expression levels. *PpARF8* was found to be highly similar to its homologs in other species (Figure S1). A review of the literature revealed that the *SlARF6* gene has a high homology with *PpARF8*. Moreover, *SlARF6* is involved in regulating the ripening process of tomato fruits and has a similar function to the Arabidopsis *AtARF8* gene [43]. In tomato plants, the formation and ripening of transgenic tomato fruits were greatly affected by the downregulated expression of *SlARF6* [44]. These reports indicated that there is a possible relationship between PpARF8 and the ripening and softening process of peach fruits.

The gene function of PpARF8 was further characterized by transient transformation in peach and tomato fruit. Overexpressed *PpARF8* was transiently transformed into peach fruit tissues, and the expression of auxin-responsive genes and genes related to fruit

ripening was analyzed (Figure 4). Aux/IAA is both an upstream and downstream gene of ARFs [45]. When *PpARF8* was overexpressed, the gene expression of *PpIAA5* was more than 1000-fold higher than that of the control group. We hypothesize that *PpARF8* has a more pronounced regulatory effect on the relative expression of *PpIAA5* and that the two may have a mutually restraining relationship. After the transient transformation of overexpressed and silenced *PpARF8* in tomato fruits, the observation of phenotypes demonstrated that *PpARF8* can promote tomato fruit ripening (Figure 5). Our findings suggest that *PpARF8* may be a positive regulator of peach fruit ripening, which will need to be further explored in the peach transformation system. For the target key enzyme regulated by PpARF8, transcriptome sequencing results showed that the expression of PpACO1, a gene related to ethylene synthesis, tended to increase with peach fruit ripening. However, PpARF8 failed to activate the expression of the downstream target gene PpACO1, and other enzymes related to cell wall degradation may be the target genes of PpARF8, which needs further in-depth study.

## 5. Conclusions

PpIAA5 may negatively regulate peach fruit softening, while PpARF8 may positively regulate peach fruit softening. Overexpression of *PpIAA5* significantly represses the expression of peach fruit ripening- and softening-related genes *PpPG* and *PpACO1* in peach fruit tissues using transient transformation. Transient overexpression of *PpARF8* in peach fruit tissues promoted the expression of *PpPA*, *PpPG*, and *PpACO1*. In addition, we identified the presence of an interaction between PpARF8 and PpIAA5. The PpIAA5-ARF8 module regulates fruit ripening and softening in peaches. These findings provide a theoretical basis for elucidating the mechanisms by which auxin signaling components regulate peach fruit ripening and softening.

**Supplementary Materials:** The following supporting information can be downloaded at: https://www.mdpi.com/article/10.3390/horticulturae9101149/s1, Table S1: Primers used for reverse transcription quantitative PCR analysis; Table S2: Primers used for vector construction; Table S3: Primers for yeast-two hybrid vector construction; Table S4: G2-vs-G1-all.gene; Table S5: G3-vs-G1-all.gene; Table S6: G4-vs-G1-all.gene; Table S7: G5-vs-G1-all.gene; Figure S1: Analysis of PpARF8 characteristics.

**Author Contributions:** Writing—original draft, Y.Q.; methodology, W.W.; formal analysis, M.C.; resources, H.Y.; software, F.Y.; writing—review and editing, Y.L. All authors have read and agreed to the published version of the manuscript.

**Funding:** This work was supported by the Beijing Municipal Natural Science Foundation (No. 6182003).

**Data Availability Statement:** Not applicable.

**Conflicts of Interest:** The authors declare no conflict of interest.

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
