# Peer review of "The PpIAA5-ARF8 Module Regulates Fruit Ripening and Softening in Peach"

_horticulturae, doi:10.3390/horticulturae9101149_

Round 1

Reviewer 1 Report

Rev 1. Major

The article presents results pertaining to the functions of the ARF8 gene in peach fruit ripening. Various practical approaches such as qRT-PCR, bimolecular fluorescence, yeast two-hybrid, and transient transformation are employed. The study is well-planned and offers compelling results that support the conclusions. However, certain aspects should be improved:

1. The use of the abbreviation "Aux/IAA" is somewhat ambiguous. In certain instances, it is used to refer to the hormone, while in others, it pertains to genes or proteins. It would be clearer to use "IAA" specifically for the auxin - indole-3-acetic acid and to abbreviate "auxin" when referring to this group of hormones more generally. When using "Aux/IAA," please clarify whether you are referencing proteins or genes.

2. Within the "Materials and Methods" section:

-Clarify the purpose behind using the pTRV vector and highlight the differences between pTRV1 and pTRV2.

-For all vectors utilized, please provide information about suppliers and/or citations from original articles.

-The primer set for TEF2 appears to be missing in Table S1. In addition, please consider adding the related the amplicons' sizes to all Tables.

-Give details about the reaction mixture used in qPCR.

-Describe how the statistical data is presented in manuscript. Were there standard deviations or errors present?

3. In the "Results" section:

-Section 3.1: While the transcriptome analysis is referenced and RNA-seq results are depicted in Fig. 1, there is no mention of the original source of these data. As per the journal requirements, the accessions/or the sequences of the gleaned transcripts/genes as well as the used transcriptome in this study should be publicly available.

-Section 3.4: There is no mention of the phylogenetic tree construction method.

- Section 3.5: The legend for Fig. 5 does not clearly indicate where the overexpressing and silencing constructs were employed. This ambiguity is exacerbated by the omission of the pTRV vectors' description in the "Materials and Methods" section. Furthermore, there's a mention of pTRV1 here, which isn't addressed in the "Materials and Methods".

4. The PpARF8 gene was transiently transformed in both peach and tomato fruits. It's unclear why only tomato phenotypes are showcased.

Reviewer 2 Report

This is a great paper.

L266. Please indicate in Material and Methods the use of the evolutionary tree built by the MEGA 5.0 software

L386. Delete in summary

Reviewer 3 Report

Dear Editor, the ms was well prepared and discussed. A minor revision is required (see my comments in the attached file).

Round 2

Reviewer 1 Report

I have now reviewed the revised version of the manuscript entitled "The PpIAA5-ARF8 module regulates fruit ripening and softening in peach". I am pleased to note that the authors have addressed all the concerns and suggestions that I previously raised.

In light of the changes made, I believe the manuscript has improved and is now suitable for publication in Horticulturae. 

However, several methodological errors still persist.

-In line 194, the authors mentioned that the pTRV1 vector was used as a control. This is not accurate. TRV is a bipartite virus, and for VIGS, two distinct A. tumefaciens strains are employed. One carries pTRV1, which encodes the viral replication and movement functions, while the other, pTRV2, contains the coat protein and the sequence used for VIGS.

-Line 205. P-values should be italicized here and in subsequent instances.
